# Programmed Cell Death Ligand 1 Expression in Circulating Tumor Cells as a Predictor of Treatment Response in Patients with Urothelial Carcinoma

**DOI:** 10.3390/biology10070674

**Published:** 2021-07-16

**Authors:** Pei-Jhang Chiang, Ting Xu, Tai-Lung Cha, Yi-Ta Tsai, Shu-Yu Liu, Sheng-Tang Wu, En Meng, Chih-Wei Tsao, Chien-Chang Kao, Chin-Li Chen, Guang-Huan Sun, Dah-Shyong Yu, Sun-Yran Chang, Ming-Hsin Yang

**Affiliations:** 1Division of Urology, Department of Surgery, Tri-Service General Hospital, National Defense Medical Center, Taipei 11490, Taiwan; peijhang@gmail.com (P.-J.C.); tailung@gmail.com (T.-L.C.); feg81230@gmail.com (Y.-T.T.); doc20283@gmail.com (S.-T.W.); en.meng@gmail.com (E.M.); weisurger@gmail.com (C.-W.T.); guman2011@gmail.com (C.-C.K.); j0921713355@yahoo.com.tw (C.-L.C.); ghsun1@gmail.com (G.-H.S.); yuds45@gmail.com (D.-S.Y.); serrina@mail.ndmctsgh.edu.tw (S.-Y.C.); 2Graduate School of Medical Sciences, National Defense Medical Center, Taipei 11490, Taiwan; xuting791107@gmail.com (T.X.); ayufish.candy@gmail.com (S.-Y.L.)

**Keywords:** urothelial carcinoma, circulating tumor cells, PD-L1, PD-L1 inhibitors

## Abstract

**Simple Summary:**

Programmed cell death ligand 1 (PD-L1) inhibitors are commonly used in treating advanced-stage urothelial carcinoma. Contrary to evaluating PD-L1 expression in tumor biopsy samples, this study assessed whether PD-L1 expression in circulating tumor cells (CTCs) can be a predictor of treatment response to PD-L1 inhibitors. The current study proved that there was no statistically significant correlation between the presence of PD-L1-positive CTCs and PD-L1 expression in tumor tissues. Moreover, PD-L1-positive CTCs at baseline could be used as a biomarker to identify patients suitable for PD-L1 blockade therapy. Dynamic changes in PD-L1-positive CTCs during the course of treatment are predictive factors of immunotherapy response and prognostic factors of disease control.

**Abstract:**

Programmed cell death ligand 1 (PD-L1) inhibitors are commonly used in treating advanced-stage urothelial carcinoma (UC). Therefore, this study evaluated the relationship between PD-L1 expression in circulating tumor cells (CTCs) and treatment response to PD-L1 inhibitors using blood samples collected from patients with UC (*n* = 23). Subsequently, PD-L1 expression and its clinical correlation were analyzed. All patients had CTCs before PD-L1 inhibitory treatment, of which 15 had PD-L1-positive CTCs. However, PD-L1-positive expression in CTCs was not correlated with PD-L1 expression in tumor biopsy samples. Patients with PD-L1-positive CTCs had better disease control (DC) rates than those without PD-L1-positive CTCs. Moreover, changes in the proportion of PD-L1-positive CTCs were associated with disease outcomes. Furthermore, the PD-L1-positive CTC count in 9 of 11 patients who achieved DC had significantly decreased (*p* = 0.01). In four patients with progressive disease, this was higher or did not change. PD-L1-positive CTCs at baseline could be used as a biomarker to identify patients suitable for PD-L1 blockade therapy. Dynamic changes in PD-L1-positive CTCs during the course of treatment are predictive factors of immunotherapy response and prognostic factors of disease control. Hence, PD-L1-positive CTCs could be employed as a real-time molecular biomarker for individualized immunotherapy.

## 1. Introduction

Advanced-stage urothelial carcinoma (UC) is an aggressive type of cancer with poor prognosis. Recently, with the discovery of novel immune-modulating agents, survival outcomes have improved in some patients. Inhibitors of the interaction between programmed cell death ligand 1 (PD-L1) and its receptor programmed cell death-1 (PD-1) has been a breakthrough revolution in the treatment of UC [1]. The use of different PD-1/PD-L1 blockers as first-line therapy for advanced-stage UC and the following treatment setting has been approved. Interestingly, patients with low PD-L1-expression tumors exhibited therapeutic effects. However, the association between PD-L1 blockade therapy and PD-L1 expression in tumor or immune cells has not been established [2].

The identification of a patient population who can benefit from these therapies has been another issue. Biomarkers that can be used to determine these patients have not been determined. Currently, several biomarkers that are better in predicting treatment responders to immunotherapy have been developed [3,4]. The predictive value of PD-L1 via immunohistochemistry (IHC) remains unknown, since a single tissue biopsy is not representative of PD-L1 expression heterogeneity, as seen in patients diagnosed with UC [5,6].

Circulating tumor cells (CTCs) in primary or metastatic tumors subsequently travel via the circulation to distant organs, leading to the formation of distant secondary tumors [7,8]. CTCs originate from different sites. Hence, the assessment of CTCs is better than tumor biopsies in identifying intratumoral heterogeneity [7,9]. Moreover, conducting a serial analysis of CTCs is possible using minimally invasive liquid biopsies to provide real-time data about tumoral changes [7,9]. Additionally, PD-L1 expression in CTCs could be used to monitor and evaluate the efficacy of immune checkpoint inhibitors [10,11,12].

The current study aimed to establish a workflow for detecting PD-L1 expression in CTCs and to analyze its relationship to the prognosis, clinicopathological features, and treatment efficacy in advanced-stage UC. This in turn can help identify biomarkers correlated with immune check point blockade therapy via liquid biopsy.

We hypothesized that CTCs obtained from peripheral blood samples via liquid biopsies could be used in not only preclinical drug development as a minimally invasive and repetitive source, but also the establishment of patient-specific platforms to identify the most effective treatment modalities.

## 2. Materials and Methods

### 2.1. Patients and Blood Sample Collection

We obtained written informed consent from all patients prior to study participation. Ethical approval to conduct this study was granted by the ethical committee of Tri-Service General Hospital (TSGHIRB NO: 2-107-05-167; date of approval: 26 January 2019). We included 10 normal individuals and 23 patients who were screened for unresectable locally advanced or metastatic UC but failed to respond to first-line platinum-based chemotherapy. Each patient provided 7.5 mL of blood, and the samples were placed in ethylenediaminetetraacetic acid (EDTA) tubes at the clinics before and 3 months following the start of second-line PD-L1 blockade therapy (1200 mg atezolizumab, administered intravenously on day 1 of each 21-day cycle). Treatment response was divided into two groups: progressive disease and disease control (DC) by achieving complete response, partial response, and stable disease, according to the Response Evaluation Criteria in Solid Tumors (RECIST), version 1.1 [13]. All samples were processed within 4 h after collection and then further evaluated via CTC analysis.

### 2.2. Immunomagnetic Bead Preparation for CTC Isolation

We used the IsoFlux^TM^ system (Fluxion, South San Francisco, CA, USA), which utilizes immunomagnetic beads targeting our selected antigens in the cancer cell surface and enriches CTCs in blood samples. The original protocol for CTC enrichment using the IsoFlux^TM^ system was modified to facilitate the replacement of IsoFlux^TM^ beads with CELLection™ Epithelial Enrich Dynabeads^®^ (Thermo Fisher Scientific, Waltham, MA, USA). Dynabeads^®^ coated with a human anti-mouse IgG was utilized to enrich cells. Then, it was incubated at room temperature with an anti-epithelial cell adhesion molecule (EpCAM) antibody (Ber-EP4, Abcam; 0.02 µg antibody/µL bead suspension). After incubation, the beads were washed with phosphate-buffered saline (PBS) with 0.1% bovine serum albumin for three times and were stored at 4 °C.

### 2.3. Sample Preparation

Erythrocytes were lysed in 45 mL of Red Blood Cell Lysis Buffer (Thermo Fisher Scientific, Waltham, MA, USA) comprised of 0.01 M potassium hydrogen carbonate, 0.155 M ammonium chloride, and 0.1 mM EDTA. The mixture was then centrifuged, and the cell pellet was washed with PBS. The cells were then resuspended in Roswell Park Memorial Institute (RPMI) medium (Thermo Fisher Scientific, Waltham, MA, USA) with 1% FBS, 1 mM CaCl_2_, and 5 mM MgCl_2_, followed by antibody-coated beads according to the original blood volume. Cells were then incubated with magnetic beads at 4 °C.

### 2.4. Sample Isolation and Collection

Beads holding the cancer cells were retrieved based on the enrichment protocol using the IsoFlux™ machine (Fluxion). Isolated cells originally held by the beads were recovered using 200 µL of RPMI medium containing 1% FBS, 5 mM MgCl_2_, and 1 mM CaCl_2_ and placed in a microcentrifuge tube (Thermo Fisher Scientific, Waltham, MA, USA). To remove the supernatant, cells attached to the beads were drawn to the bottom of the tube using a magnet. Then, the cells were fixed in 4% PFA and placed onto glass slides. Next, a circle corresponding to the size of the magnet was drawn on the glass slide using a water repellent Dako pen (Agilent, Santa Clara, CA, USA). The magnet was placed below the glass slide when adding or removing the buffer from the cells.

### 2.5. CTC Analysis and Immunofluorescence

Isolated cells were mounted and fixed on slides and inhibited for 5 min in 10% normal donkey serum and subsequently stained with PE-conjugated anti-CD45 (5B-1) antibody (1:200; MACS Miltenyi Biotec, Bergisch Gladbach, Germany). Then, they were permeabilized using 0.2% Triton X-100 in PBS containing 0.5% BSA and 2 mM EDTA and stained with FITC-conjugated anti-cytokeratin (CK3-6H5) antibody (1:10; MACS Miltenyi Biotec, Bergisch Gladbach, Germany) and Alexa Fluor^®^ 647-conjugated anti-PD-L1 antibody [SP142] (1:100; Abcam, New Taipei City, Taiwan). To stain the cell nuclei, cells were incubated in DAPI afterwards. For mounting, Dako Faramount Aqueous Mounting Medium (Agilent, Santa Clara, CA, USA) was used. Images were captured using a fluorescent microscope (Axio Scan.Z1, Zeiss, Oberkochen, Germany). CTCs were categorized into CK-positive, CD45-negative, and nucleated.

### 2.6. Cell Culture and Spike-In Experiment

The MB49, T24, and J82 cells were procured from the Bioresource Collection and Research Center (Taiwan). They were cultured in culture media as per American Type Culture Collection recommendations and incubated at 37℃ in an incubator with a CO_2_ of 5% and humidity of 95%. Before the spike-in experiments, cells were collected using trypsin–EDTA (Sigma-Aldrich, St. Louis, MO, USA) and were resuspended in RPMI medium. The cell concentration was manually measured by counting the viable cells in Trypan Blue Solution (Sigma-Aldrich, St. Louis, MO, USA) at a cell–solution ratio of 1:1. The suspensions were subsequently spiked into 7.5 mL of whole blood for the enrichment process using the IsoFlux^TM^ system.

### 2.7. Immunocytochemistry for PD-L1

Tumor tissues were collected from the participants (*n* = 23). Formalin-fixed paraffin-embedded tumor blocks were sliced into 3-μm thick sections. Tumor PD-L1 expression was measured using the Ventana PD-L1 (SP142) assay (Ventana Medical Systems, Tucson, AZ, USA), and PD-L1 positivity was defined as ≥5% of tumor-infiltrating immune cells via membrane staining.

### 2.8. Statistical Analysis

All experiments were conducted at least three times unless stated otherwise. Continuous variables were expressed as mean ± SD. The Mann–Whitney U test was used to compare the differences between groups. Data were analyzed using Prism version 9 (GraphPad Software Inc., San Diego, CA, USA). *p*-values of <0.05 were considered significant.

## 3. Results

### 3.1. Detection of Circulating UC Cells

The isolation of metastatic CTCs from cancer patients remained a challenge due to the rarity of CTCs and lack of specific markers. Thus, to examine the CTC capture efficiency, we used the IsoFlux^TM^ platform to capture human UC cell lines. Using the EpCAM antibody, the capture rates for MB49, J82, and T24 cells were 89.8%, 89.2%, and 88.1%, respectively (Figure 1A). We further tested our platform on 10 normal individuals. No CTCs were found in their peripheral blood samples (Figure 1B). Next, we used the platform to evaluate samples from UC patients. Participant characteristics are depicted in Table 1. None of the participants had other cancers concurrent with UC. All patients had metastatic UC and received anti-PD-L1 therapy (atezolizumab). Their mean age at diagnosis was 59.7 ± 5.6 years. Among them, 11 (48%) presented with visceral metastases, six (26%) with liver metastases, five (22%) with lymph nodal metastases, and one (4%) with brain metastasis. The mean follow-up time from the start of PD-L1 inhibition therapies was 6.6 ± 2.2 months. CTCs were enriched and detected in all patients after immunofluorescence staining. Cancer cells were evaluated based on the morphology and expression of CK (Figure 2). We identified CTCs in all patients with UC prior to treatment initiation (Figure 1B; Table 2; mean CTCs/7.5 mL = 7.2 ± 4.8). The same patients were evaluated during the post-treatment period (mean CTCs/7.5 mL = 6.3 ± 3.8).

### 3.2. Detection of PD-L1-Positive CTCs and Its Related Clinical Outcomes

We further examined the PD-L1 status of CTCs isolated using the IsoFlux^TM^ system. PD-L1-positive CTCs in the blood samples were quantified, and the expression of PD-L1 in CTCs was evaluated. PD-L1 was located in both the cell cytoplasm and cell membrane. Before treatment, all patients had CTCs, ranging from two to 16 (median: six). Moreover, 15 of 23 patients (65%) had PD-L1-positive CTCs, ranging from one to seven (median: three).

The patients’ clinical outcomes were assessed based on PD-L1-positive CTC status. Patients were categorized into two groups: those with progressive disease and those with DC (i.e., an ongoing complete response, partial response, or stable disease) as the initial response. Based on RECIST criteria, 12 (52%) patients responded to the treatment, and 11 (48%) developed tumor progression (Table 3). The median follow-up period was 7 (range: 4–13) months. After 3 months of treatment, 11 of the 15 patients with PD-L1-positive CTCs (73%) achieved DC, and four (27%) had progressive disease. Among the patients without PD-L1-positive CTCs, only one (12.5%) achieved DC, and seven (87.5%) had progressive disease 3 months after treatment. Initially, there was no significant correlation between clinical response to treatment and baseline total CTC (*p* = 0.77) or baseline PD-L1-positive CTC count (*p* = 0.24) (Figure 3). Interestingly, no correlation was found between treatment response and age (*p* = 0.17), sex (*p* = 0.83; chi-squared test), or primary tumor sites (*p* = 0.15; bladder vs. upper urinary tract; chi-squared test).

All peripheral blood samples were prospectively collected longitudinally. We assessed the dynamics of total CTC and PD-L1-positive CTC count among patients who had PD-L1-positive CTCs during PD-L1 blockade therapy. The total CTC count in five of 12 patients (42%) with DC and five of 11 patients (45%) with progressive disease decreased. Despite the presence of PD-L1-positive CTCs, there was no correlation between changes in total CTC count and clinical response to PD-L1 blockade therapy (Table 3 and Figure 4). Furthermore, the PD-L1-positive CTC count in nine of 11 patients (82%) with DC significantly decreased at the 3-month follow-up (*p* = 0.01) (Table 3; Figure 5A). Meanwhile, in four patients (100%) with progressive disease, this increased or did not change (Figure 5B).

### 3.3. Comparison of PD-L1 Expression in CTCs and Tumor Biopsies

During the initial diagnosis, we collected primary tumor samples of 23 CTC-positive patients. PD-L1 expression was assessed via primary tumor biopsy during the initial diagnosis (Figure 6). In total, 15 patients (65%) had low PD-L1 expression, whereas eight (35%) had high expression. Consistent PD-L1 expression in CTCs, and its corresponding tumor biopsy samples, was observed in only 10 patients (43%).

Inconsistent PD-L1 expression in CTCs and matched tumor specimens was observed in 13 patients. Of these 13 patients, three (23%) had low PD-L1 expression in CTCs but high expression in tumor biopsy samples. Meanwhile, the other 10 patients (77%) had high PD-L1 expression in CTCs but low expression in tumor biopsy samples.

## 4. Discussion

UC is characterized by remarkable intratumoral heterogeneity, and it has become evident that it is caused by not only genetically distinct subclones, but also phenotypic heterogeneity and plasticity in each subclone [14,15]. CTCs in the peripheral circulation represent a readily accessible liquid biopsy, and it is a reliable tool for predicting prognosis, monitoring treatment response, and representing intratumoral heterogeneity in different types of solid cancers [9,16,17,18,19,20,21,22,23,24]. The detection of CTCs in cancers has been widely assessed. Moreover, research about CTCs in UC has progressed [9,25,26,27]. In literature, EpCAM is the main molecule for capturing CTC [7,9,28]. The CellSearch^®^ system, the only the U.S. Food and Drug Administration-approved CTC detection method, is EpCAM-dependent. In our study, peripheral blood CTCs from patients with UC were enriched using the IsoFlux^TM^ platform with CELLection™ Dynabeads^®^, which also utilized the immunomagnetic capture of CTCs based on EpCAM expression. Alva et al. [29] demonstrated the higher sensitivity of the IsoFlux^TM^ method compared with CellSearch^®^ and its facilitation of cell molecular profiling via next-generation sequencing. A DNA linker helps antibodies to attach to the surface of the Dynabeads. This linker provides a cleavable site to release and remove the beads from the cells after isolation, which otherwise would not be possible with the IsoFlux beads.

Goldkorn et al. [30] found that baseline total CTC count could be a valuable predictive marker for identifying patients with metastatic castrate-sensitive prostate cancer that may respond favorably to hormonal therapies, from those that may benefit from alternative timely interventions. Nevertheless, Coumans et al. [31] revealed changes in the total CTC count, which were correlated with overall survival. Moreover, the study showed that treatment should aim to eliminate all CTCs. However, no correlation was observed between clinical responses to PD-L1 blockade therapy and baseline total CTC count or fluctuations in total CTC count. In addition to the small sample size of our study, this discrepancy might be attributed to the antibodies used in different studies. Moreover, it could be caused by the small subgroup of more aggressive CTCs. However, identifying these minor clones is challenging because of the limited markers chosen.

Previous studies in the field have shown that PD-L1-positive CTCs were considered a reliable surveillance tool for treatment response analysis among patients with lung cancer [10,32,33]. In patients diagnosed with head and neck or breast cancer, PD-L1-positive CTCs have a prognostic predictive value [34,35]. In this study, only one of eight patients (12.5%) without PD-L1-positive CTCs slightly responded to PD-L1 inhibitor therapy, whereas the rest had tumor progression. Thus, patients with UC not harboring PD-L1-positive CTCs did not have a favorable response to PD-L1 blockade therapy. Clinicians should monitor PD-L1 expression status in CTCs prior to treatment commencement.

Importantly, our findings revealed that ongoing changes in PD-L1-positive CTCs, not baseline PD-L1-positive CTC count, influence the response to PD-L1 blockade therapy. A decrease in PD-L1-positive CTC count is a good predictive factor for prognosis and oncological response to PD-L1 blockade therapy. During treatment, dynamic changes in PD-L1 expression in primary tumor cells may occur, resulting in different outcomes. However, this could be missed in a single tumor biopsy. For instance, one participant in our study, a 53-year-old female patient with metastatic bladder UC and high PD-L1 expression in CTCs but low expression in primary tumor, responded to PD-L1 inhibition and achieved a clinically complete response. No PD-L1-positive CTC was found in the peripheral blood sample after four cycles of atezolizumab. Hence, analyzing PD-L1 expression in primary tumors has only partial predictive value. Furthermore, performing serial tumor biopsies is not practical. Thus, CTCs analysis can be used as an alternative to minimally invasive procedures. The assessment of dynamic PD-L1 changes in CTCs could monitor real-time tumor changes resulting from PD-L1 blockade therapy. This helps clinicians identify individuals who may benefit from PD-L1 blockade therapy and predict therapeutic responses. In addition, the ongoing presence of PD-L1-positive CTCs might reflect resistance adopted by cancer cells. Our findings may support the theory that a certain group of PD-L1-positive CTCs (e.g., circulating cancer stem cells [CSCs]) could be a prognostic biomarker for immunotherapies in UC. Meanwhile, CTCs isolated from peripheral blood are highly dependent on the markers we chose, and some groups of cells might be missed. These circulating CSCs may be resistant to PD-L1 blockade therapy, and they were not eliminated [36]. Thus, some PD-L1-positive CTCs patients did not respond to treatment. The PD-L1-positive CTCs could provide information about how cancer cells gain resistance to PD-L1 blockade therapy, because they can be assessed through serial liquid biopsies. Nevertheless, the theory is currently not understood. Thus, further studies with longer follow-up and larger cohorts must be performed to address this point. In line with previous reports about other types of cancer, we observed that the PD-L1 expression in primary tumor specimens had no bearing on PD-L1 expression in CTCs [35,37,38]. This further suggests intratumoral heterogeneity, with respect to PD-L1 expression in primary tumors and CTCs. Furthermore, the PD-L1 status of primary tumors had no impact on a patient’s overall survival, a finding consistent with the previous analysis performed by Ghate et al. [39].

Apart from PD-L1 CTCs, other factors (e.g., sex and age) may play a role in cancer treatment. In the literature, various studies support a male-favored benefit in checkpoint inhibitor therapy. Overall survival and progression-free survival demonstrate different trends in these analyses for males and females. These can result from genetic and hormonal differences between the sexes that diversely affect immune responses to immunotherapy [40,41,42]. However, the efficacy of immunotherapy appears to differ between older and younger age groups in terms of the types of cancer, some reporting no difference and others reporting greater or low efficacy in the elderly age group [43,44]. However, in our cohort, no correlation was found between the response to immunotherapy and sex, age, or primary tumor sites.

The identification of PD-L1-positive CTCs from liquid biopsy samples may be a promising means to evaluate PD-L1 expression patterns in UC, regardless of the heterogeneous expression in the primary tumor site and the metastatic lesions. In the treatment of cancer, breakthrough therapy with checkpoint inhibitors may become more important in the near future. Nevertheless, reliable predictive treatment-response biomarkers for patient selection must be identified.

The current study had several limitations. The small sample size limited our conclusions concerning the clinical application of PD-L1 expressions in CTCs. Hence, randomized clinical trials with larger cohorts must be conducted to validate the results of this research. Furthermore, standardized approaches for PD-L1-positive CTCs with a high purity and specificity and valid reproducibility, allowing comparability between different systems, are significantly important.

## 5. Conclusions

CTCs are characteristic features of advanced or metastatic UC. However, we found that the presence of PD-L1-positive CTCs was not related to PD-L1 expression in tumor tissues. Additionally, neither baseline total CTC nor fluctuations in total CTC count was associated with the response to PD-L1 blockade therapy. The majority of patients without PD-L1 expression in CTCs were unresponsive to PD-L1 blockade therapy. Therefore, PD-L1-positive CTCs at baseline could be used as a biomarker to identify patients suitable for PD-L1 blockade therapy. Nevertheless, in the absence of a correlation between oncologic responses and PD-L1-positive CTCs in patients with UC, PD-L1-positive CTCs cannot be used as a prognostic marker. Dynamic changes in PD-L1-positive CTCs during treatment course are predictive factors of response to immunotherapy and prognostic factors of disease control.

## Figures and Tables

**Figure 1 biology-10-00674-f001:**
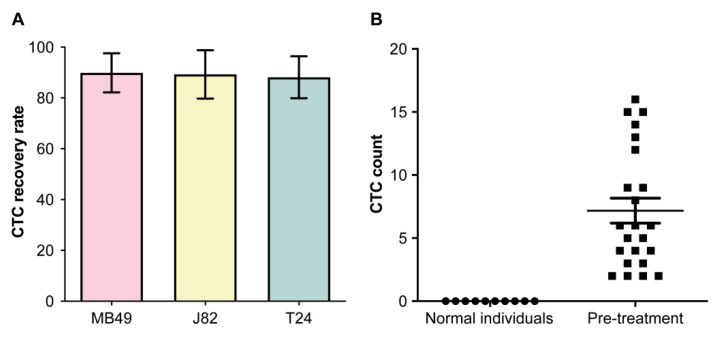
(**A**) Recovery rate of urothelial carcinoma (UC) cell lines using the IsoFlux system (*n* = 8 for each cell line). (**B**) Circulating tumor cell (CTC) count in normal individuals and UC patients at baseline.

**Figure 2 biology-10-00674-f002:**
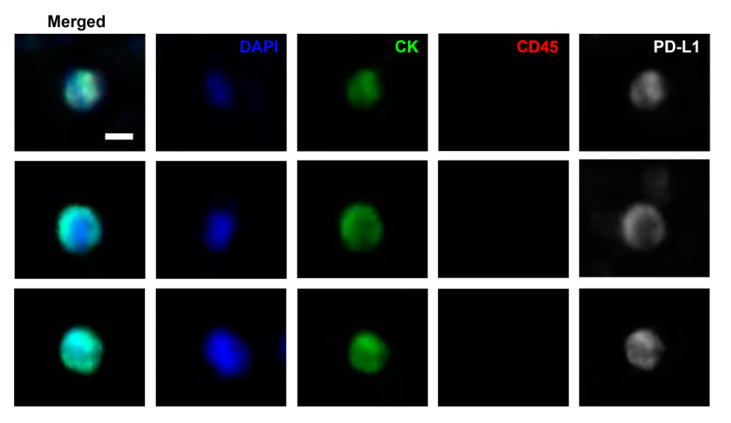
Immunofluorescence staining of representative programmed cell death ligand 1 (PD-L1, white) circulating cancer cells obtained using the IsoFlux system. Cancer cells fulfilled the criteria for circulating tumor cells, including nucleated (blue), CK-positive (green), and CD45-negative (non-red) cells. Scale bar, 5 µm.

**Figure 3 biology-10-00674-f003:**
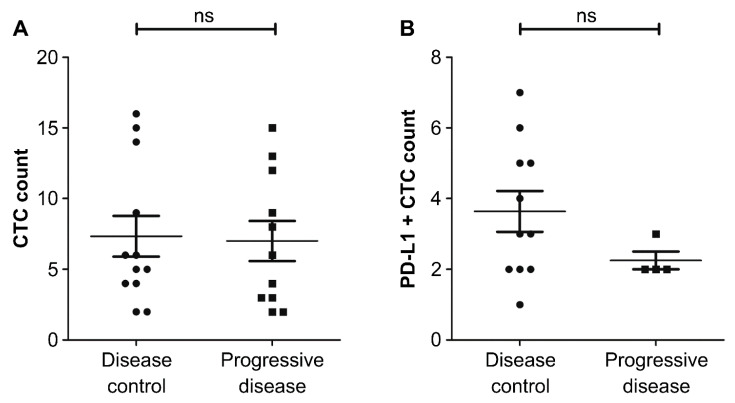
(**A**) Baseline circulating tumor cell (CTC) count of patients with urothelial carcinoma (UC) in the disease control group and the progressive disease group (*p* = 0.78). (**B**) Baseline programmed cell death ligand 1 (PD-L1)-positive CTC count of patients with UC in the disease control group and the progressive disease group (*p* = 0.22). ns: not significant.

**Figure 4 biology-10-00674-f004:**
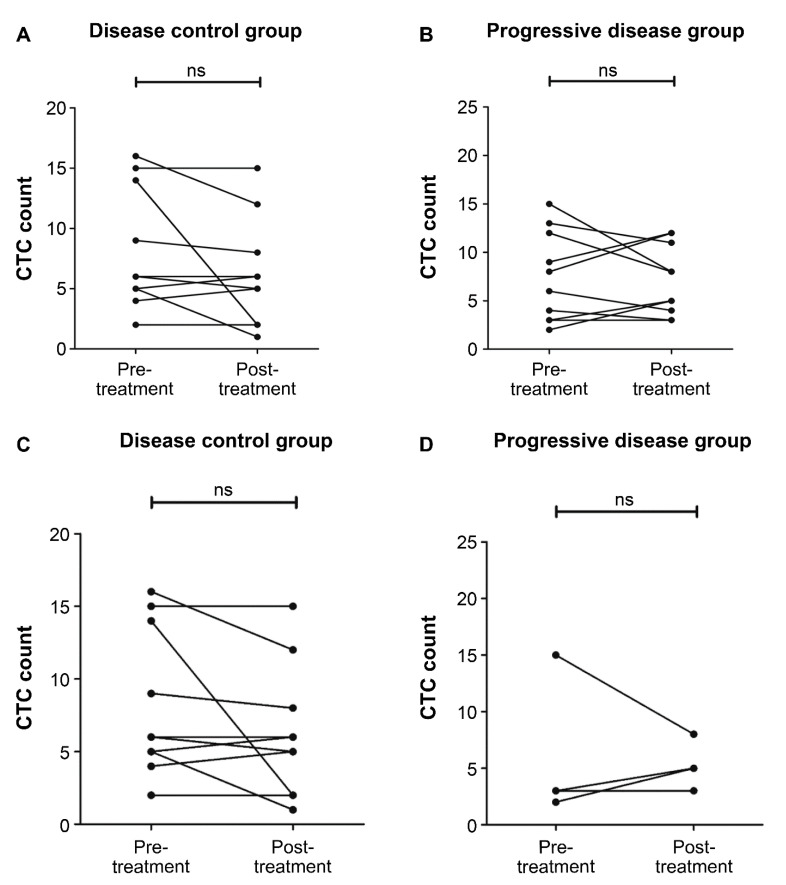
Time course analysis of total circulating tumor cell (CTC) count from blood samples of patients with urothelial carcinoma (UC). The CTC levels were evaluated before and 3 months after the start of programmed cell death ligand 1 (PD-L1) blockade therapy. (**A**) Total CTC count in the disease control group (*p* = 0.22). (**B**) Total CTC count in the progressive disease group (*p* = 0.88). (**C**) Total CTC count in the disease control group of patients with PD-L1-positive CTCs (*p* = 0.14). (**D**) Total CTC count in the progressive disease group of patients with PD-L1-positive CTCs (*p* = 1.00). ns: not significant.

**Figure 5 biology-10-00674-f005:**
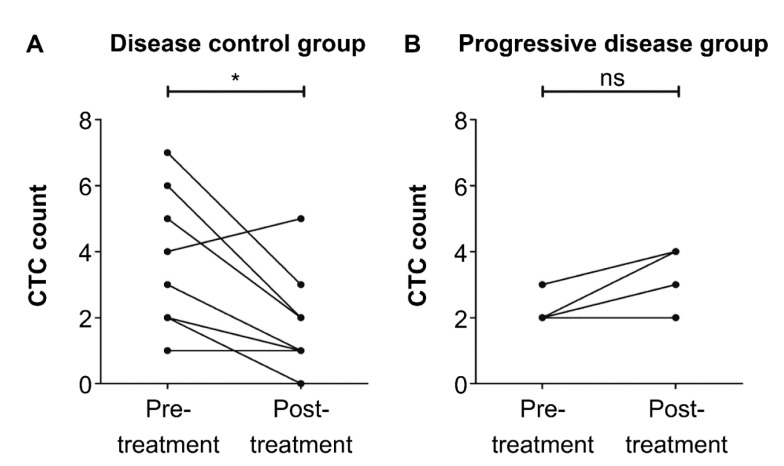
Time-course analysis of programmed cell death ligand 1 (PD-L1)-positive circulating tumor cell (CTC) count in patients with urothelial carcinoma (UC). The CTC levels were evaluated before and 3 months following the start of PD-L1 blockade therapy. (**A**) PD-L1-positive CTC in the disease control group of patients with PD-L1-positive CTCs (* *p* = 0.01). (**B**) PD-L1-positive CTC in the progressive disease group of patients with PD-L1-positive CTCs (*p* = 0.25). ns: not significant.

**Figure 6 biology-10-00674-f006:**
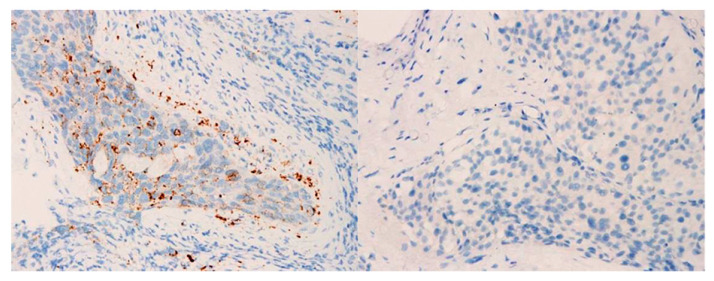
Representative immunohistochemistry images of programmed cell death ligand 1 in the primary samples collected from patients with urothelial carcinoma. Magnification: 400×. **Left**: immune cells (IC) = 15 (high expression); **right**: IC = 0 (low expression).

**Table 1 biology-10-00674-t001:** Demographic and clinical features of patients with urothelial carcinoma.

	Mean ± SD	No. of Patients	%
Age	59.7 ± 5.6	23	100
Sex			
Male		11	48
Female		12	52
Tumor location			
Bladder		12	52
Kidney		6	26
Ureter		5	22
Metastatic sites			
Visceral		11	48
Liver		6	26
Lymph nodes		5	22
Brain		1	4
Follow-up duration (months)	6.6 ± 2.2		

**Table 2 biology-10-00674-t002:** Enriched baseline circulating tumor cell (CTC) count.

CTC Populations	No. of Patients	Pre-Treatment Count/7.5 mL (Mean ± SD)	Post-Treatment Count/7.5 mL (Mean ± SD)	*p*-Value
Total CTC count	23	7.2 ± 4.8	6.3 ± 3.8	0.63
PD-L1-positive CTC count	15 (65%)	3.3 ± 1.8	2.1 ± 1.4	0.06

**Table 3 biology-10-00674-t003:** Circulating tumor cell (CTC) count and overall response rates to programmed cell death ligand 1 (PD-L1) blockade therapy.

Overall Responses	No. of Patients	Pre-Treatment CTC Count/ 7.5 mL (Mean ± SD)	Post-Treatment CTC Count/ 7.5 mL (Mean ± SD)	*p* Value	Pre-Treatment PD-L1-Positive CTC Count/ 7.5 mL (Mean ± SD)	Post-Treatment PD-L1-Positive CTC Count/ 7.5 mL (Mean ± SD)	*p* Value
**All Patients**							
Disease control, n (%)	12 (52)	7.3 ± 5.0	5.8 ± 4.2	0.22	-	-	-
Progressive disease, n (%)	11 (48)	7.0 ± 4.7	6.9 ± 3.5	0.88	-	-	-
**Patients with** **PD-L1-Positive CTCs**							
Disease control, n (%)	11 (73)	7.6 ± 5.1	5.8 ± 4.4	0.14	3.6 ± 1.9	1.7 ± 1.3	0.01
Progressive disease, n (%)	4 (27)	5.8 ± 6.2	5.3 ± 2.1	1.00	2.3 ± 0.5	3.3 ± 1.0	0.17
**Patients without** **PD-L1-Positive CTCs**							
Disease control, n (%)	1 (12.5)	4	5	-	-	-	-
Progressive disease, n (%)	7 (87.5)	7.7 ± 4.0	7.9 ± 3.9	1.00	-	-	-

## Data Availability

The data presented in this study are available on request from the corresponding author. The data are not publicly available due to privacy reasons.

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
