# Peer review of "Programmed Cell Death Ligand 1 Expression in Circulating Tumor Cells as a Predictor of Treatment Response in Patients with Urothelial Carcinoma"

_biology, 2021, doi:10.3390/biology10070674_

Round 1

Reviewer 1 Report

The author tried to use the IsoFluxTM system to detect the PD-L1-positive CTCs in urothelial carcinoma (UC). The study proved that there was no correlation between the presence of PD-L1-positive CTCs and PD-L1 expression in tumor tissues, and tried to prove the CTC count could be a prognostic biomarker for PD-1/PD-L1 blockade immunotherapy in UC. However, the study seems exist some problems.

1) There was no control group in Fig.1 to exclude the capture rates of EpCAM antibody for other tumor or normal cells.

2) The correlation between the presence of PD-L1-positive CTCs and PD-L1 expression in tumor tissues could be an independent result.

3) The authors thought the antibodies used in different studies could be one of reasons for discrepancy in the correlation between clinical responses to PD-L1 blockade therapy and baseline total CTC count and changes in total CTC count. Please add some corresponding experiments to prove this conjecture.

4) The author thought their findings supported the theory “a certain group of PD-L1-positive CTCs (e.g., circulating cancer stem cells [CSCs]) could be a prognostic biomarker for immunotherapies in UC”. However, judging from the results, there may be insufficient data to prove this point.

5) The data was used repeatedly. Some results were explained in both tables and figures.

Reviewer 2 Report

  • Methods must be improved.
  • Elaborate more about spike-in experiment and its purpose.
  • P-value should be mentioned in the legends of all figures.
  • Correction in header (from 2020 to 2021).
  • Results can be improved.
  • Discussion must be improved.
  • Number of replicates should be more than 8 in each experiment

Reviewer 3 Report

The manuscript entitled “Programmed Cell Death Ligand 1 Expression in Circulating Tumor Cells as a Predictor of Treatment Response in Patients with Urothelial Carcinoma” by Chiang et al describes PD-L1-positive CTCs is a predictor of the efficacy of PD-L1 blockade therapy. In this manuscript authors have suggested that change in PD-L1-positive CTCs can be used in monitoring therapeutic response. However, some of the findings are exaggerated and needs further work to be used by researchers in the field. I do think the manuscript needs major revision before publication in Journal Biology.

Major Comments:

  1. Authors have suggested that predictive value of PD-L1 via IHC is questionable due to the heterogeneity of its expression. Though authors found PD-L1+ CTCs using IHC. Author’s haven’t provided the CTCs PD-L1 expression data in Urothelial patients before and after treatment as well as in progressive disease. Also, in matched healthy tissue for comparison. Given very few CTCs in blood the difference 3.3±1.8 vs 2.1±1.4 is not be very huge to suggest predictability.
  2. Does patient have received standard care.? If yes, is there any impact on CTCs count and PD-L1 expression.
  3. PD-L1 blockade therapy can be confounded by age, sex and other factors. Do authors see any such bias? All the patients received PD-L1 therapy based on severity of disease? Do these patients do not show any recovery response after SOC?
  4. It will be interesting to see the PD-L1 expression in recovering patients both by CTCs (1 to None) and available expression/literature data.
  5. Figure-5 show pretty good shift in CTCs in pre- and post-treatment, but we do see decrease in patients size like in progressive disease. Overall, it seems among 4 progressive patients only 2 shows increase in CTCs.
  6. Small sample size (n=23) is not enough to suggest the predictor value of PD-L1. However, authors can report their observations with support from literature. And provide limitations of the study cohort in manuscript.

Minor Comments

  1. Figure-1 provides IsoFlux efficacy which can be appropriate system but not strength of findings hence can be moved to supplementary.
  2. Disease control term is confusing.
  3. Figire-3B shows few progressive disease patients (n=4) with PD-L1+ CTCs but the explanation is not found.
  4. Authors have mentioned “Patients with UC not harboring PD-L1-positive CTCs do not have a favorable response to PD-L1 blockade therapy.”, it will be important to mention those data to support.

Round 2

Reviewer 1 Report

I have no other comments for the article, except for the need to modify some of the format. For example, the p-value should be marked at the end of the data described, not a new sentence; "et al" should be italicized. Please carefully review the formats.

Author Response

We thank the reviewers for the comments. We have checked the manuscript in detail and corrected the format as required. Besides, we had addressed this issue for English editing again during major revision, and we will upload the revised manuscript.

Reviewer 2 Report

Authors provided significant information to improve the quality of the manuscript including the detailed description of the figures. Results were also improved. Discussion and conclusions could be improved.

Author Response

We thank the reviewers for the comments. Indeed, there are still many shortcomings need to improve in our research. This is the goal we must strive for in our future work.

Reviewer 3 Report

Authors have replied to the major comments appropriately. Some minor editing is required otherwise I accept the manuscript.

Author Response

We thank the reviewers for the comments. We had addressed this issue for English editing again during major revision, and we will upload the revised manuscript.